Multi-benthic size approach to unveil different environmental conditions in a Mediterranean harbor area (Ancona, Adriatic Sea, Italy)

Baldrighi Elisa 1 2 elisa.baldrighi@irbim.cnr.it
http://orcid.org/0000-0001-6330-152X Pizzini Sarah 1 3
Punzo Elisa 1
Santelli Angela 1
Strafella Pierluigi 1
Scirocco Tommaso 4
Manini Elena 1
http://orcid.org/0000-0002-5847-4722 Fattorini Daniele 5 6
http://orcid.org/0000-0002-5381-6690 Vasapollo Claudio 1 claudio.vasapollo@cnr.it
1 Institute for Biological Resources and Marine Biotechnologies—IRBIM, National Research Council—CNR, Italy , Ancona, Marche , Italy
2 Department of Biology, University of Nevada-Reno , Reno, Nevada , USA
3 Fano Marine Center, The Inter-Institute Center for Research on Marine Biodiversity, Resources and Biotechnologies , Fano , Italy
4 Institute for Biological Resources and Marine Biotechnologies—IRBIM, National Research Council—CNR, Italy , Lesina , Italy
5 Dipartimento di Scienze della Vita e dell’Ambiente (Disva), Università Politecnica delle Marche (Univpm) , Ancona , Italy
6 Consorzio Nazionale Interuniversitario per le Scienze del Mare (Conisma), Unità di Ricerca di Ancona (Italy) , Ancona, Marche , Italy
Corte Guilherme
Electronic publication date: 2023 Jun 28
Publication date: 2023
Volume: 11
Electronic Location ID: e15541
Received 2023 Jan 11; Accepted 2023 May 21
Copyright: © 2023 Baldrighi et al.
Copyright year: 2023
Copyright holder: Baldrighi et al.
License: This is an open access article distributed under the terms of the Creative Commons Attribution License, which permits unrestricted use, distribution, reproduction and adaptation in any medium and for any purpose provided that it is properly attributed. For attribution, the original author(s), title, publication source (PeerJ) and either DOI or URL of the article must be cited.
License URL: https://creativecommons.org/licenses/by/4.0/

Keywords: Meiofauna, Macrofauna, Adriatic sea, Harbor, Contaminants, Benthic size, Mediterranean

Funding: IPA Adriatic Cross-Border Cooperation Program Ballast Water Management System for Adriatic Sea Protection (BALMAS) IPA Adriatic Cross-Border Cooperation Program Authorities This publication has been produced with the financial assistance of the IPA Adriatic Cross-Border Cooperation Program—strategic project Ballast Water Management System for Adriatic Sea Protection (BALMAS). The contents of this publication are the sole responsibility of authors and can under no circumstances be regarded as reflecting the position of the IPA Adriatic Cross-Border Cooperation Program Authorities. The funders had no role in study design, data collection and analysis, decision to publish, or preparation of the manuscript.

==============================
Harbors are hubs of human activity and are subject to the continuous discharge and release of industrial, agricultural, and municipal waste and contaminants. Benthic organisms are largely known to reflect environmental conditions they live in. Despite meio- and macrofauna interacting within the benthic system, they are ecologically distinct components of the benthos and as such may not necessarily respond to environmental conditions and/or disturbances in the same way. However, in a few field studies the spatial patterns of meio- and macrofauna have been simultaneously compared. In the present study, we assess the response and patterns in the abundance, diversity, and distribution of the two benthic size classes to the different environmental conditions they live in (i.e., sediment concentrations of selected trace metals and polycyclic aromatic hydrocarbons (PAHs); organic matter contents and grain size) characterizing the Ancona Harbor (Adriatic Sea). Meio- and macrofauna provided partially similar types of information depending on the indices used (univariate measures or community structure/species composition) and the different ‘response-to-stress’. The community structure (i.e., taxa composition) of both benthic size components clearly showed differences among sampling stations located from inside to outside the harbor, reflecting the marked environmental heterogeneity and disturbance typically characterizing these systems. Notwithstanding, the univariate measures (i.e., meio- and macrofauna total abundance, diversity indices and equitability) didn’t show similar spatial patterns. Meiofauna were likely to be more sensitive to the effects of environmental features and contaminants than macrofauna. Overall, trace metals and PAHs affected the community composition of the two benthic components, but only the meiofauna abundance and diversity were related to the environmental variables considered (i.e., quantity and quality of organic matter). Our results pinpoint the importance of studying both meio- and macrofauna communities, which could provide greater insight into the processes affecting the investigated area and reveal different aspects of the benthic ecosystems in response to harbor conditions.

Introduction

Coastal waters are widely recognized as marine areas of high ecological and economic value, but also as highly threatened zone, exposed to multiple human activities (e.g., harbors) and their negative impacts (Travizi et al., 2019).

Harbors are essential to the economic growth of coastal regions, where maritime traffic, shipping, international trade, and fishing are continuously increasing (Simonini et al., 2005; Franzo et al., 2022). Harbors are usually characterized by high sediment pollution levels due to heavy metals and hydrocarbons caused by intense maritime traffic and huge organic matter loads (Covazzi Harriague et al., 2007; Baldrighi et al., 2019). The high concentrations of contaminants and the relevant inputs of organic matter represent a persistent and ongoing threat, especially for the biota living in the sediment (Travizi et al., 2019; Franzo et al., 2022). In addition, the exposure of the innermost part of harbors to both wind and waves is limited and may create conditions of reduced water circulation, favoring sedimentation processes, anoxia, and trapping pollutants (Guerra-García & García-Gómez, 2004a; Spagnolo, Scarcella & Sarappa, 2011). The EU Water Framework Directive (WFD; Directive 2000/60/EC; European Communities (EC), 2000) considers harbors as ‘heavily modified water bodies’, which cannot meet the common criteria of good ecological quality status. Therefore, their effective management is crucial for the sustainable use of these maritime spaces and for the protection of the adjacent coastal habitats (Chatzinikolaou et al., 2018; Franzo et al., 2022).

Benthic organisms are largely known to reflect environmental conditions they live in. Among benthic components, species of meio- and macrofauna are widely recognized as good ecological indicators (Schratzberger et al., 2003; Patrício et al., 2012). Benthic meiofaunal and macrofaunal communities are regularly utilized in impact assessment, but very few studies are carried out taking into account both communities (Whomersley et al., 2009; Frontalini et al., 2011; Covazzi Harriague, Albertelli & Misic, 2012; Xu, Cheung & Shin, 2014). The meio- and macrofauna are ecologically distinct components of the benthos and as such may not necessarily respond to environmental conditions and/or disturbances in the same way (Schratzberger et al., 2003; Patrício et al., 2012).

Aside from differences in body size, meio- and macrobenthos each have a series of distinctive ecological and evolutionary characteristics supporting an expectation of differences in their response to environmental conditions. Meiofauna is characterized by small size, high abundance in marine sediments, ubiquitous distribution, rapid generation times (i.e., months), fast metabolic rates, and absence of a planktonic phase, resulting in a short response time and high sensitivity to different environmental conditions and certain types of disturbance (Giere, 2009). Meiofauna are often classified as permanent (species spending their whole lives as meiofauna) or temporary (animals that start off as meiofauna but grow into macrofauna) (Giere, 2009). In some environments (e.g., open slope systems), temporary meiofauna (e.g., Bivalvia, Oligochaeta, Amphipoda, Nemertea, Priapulida, Holothuroidea, Ascidiacea Cnidaria and Decapoda larvae) have been found to constitute a high percentage of the meiobenthic community (Bianchelli et al., 2010). All these organisms are expected to grow into macrofauna and become part of the macrobenthic population of the slope systems. According to modern phylogenetic approaches, seven phyla belong exclusively to meiofauna, while seventeen phyla are accommodated between meio- and macrofauna (Giere & Schratzberger, 2023).

Meiobenthic organisms cover different ecological roles according to their trophic group, living mode, locomotion adaptation to move and digging in different kind of sediment grains and they comprise both unicellular (e.g., Foraminifera, Ciliata) a metazoan organism (Giere, 2009). Moreover, because of their high density even small volumes of samples can be analyzed to assess meiofaunal changes over different spatial scales and environmental conditions (e.g., Schratzberger & Ingels, 2018; Ridall & Ingels, 2021). Consequently, meiofaunal communities have generated considerable interest as potential indicators of anthropogenic disturbances in aquatic ecosystems (e.g., Ridall & Ingels, 2021; Semprucci, Grassi & Balsamo, 2022).

There are some advantages in using macrofauna, strictly linked to the functional traits of this group: (1) relative longevity, with many species having life spans in excess of 2 yr, allows the macrofauna to integrate responses to environmental pressures over extended time periods (Simboura & Zenetos, 2002); (2) sedentary or sessile mode of life, therefore organisms are not able to escape stress and integrate the environmental quality in a given area; (3) relatively easy to sample quantitatively, even if larger amount of sediment are needed compared to the meiofauna; (4) relatively easy taxonomic identification and available taxonomic keys for most groups and (5) well-documented and predictive response to a number of environmental stressors (thus, community changes can be interpreted with a degree of confidence) (Gray et al., 1988; Somerfield et al., 2006; Todorova, Doncheva & Trifonova, 2020). Meio- and macrofauna are ecologically distinct components of the benthos, however both benthic size classes are highly influenced by some main sediment features such as the grain size (Moreno et al., 2008b; Pereira et al., 2018a), the quantity and quality of organic matter (Vezzulli et al., 2003; Papageorgiou, Kalantzi & Karakassis, 2010) and presence of pollutants (McCready, Birch & Long, 2006; Dauvin et al., 2017). Macrobenthic invertebrates have been identified by the EU WFD as key biological components to assess the ecological status of aquatic ecosystems, due to their important role in ecosystem functioning and to their involvement in food-web nutrient recycling (Punzo et al., 2017).

The complementary use of two sets of faunistic groups with contrasting ecological characteristics could provide greater insight into the processes affecting such an area (Austen, Warwick & Rosado, 1989).

Up to now, field studies where the spatial patterns of meio- and macrofauna have been simultaneously compared, changes in both assemblages as a response to natural gradients were found to be scattered across habitats (Somerfield et al., 2006; Papageorgiou, Kalantzi & Karakassis, 2010; Patrício et al., 2012). These investigations have demonstrated the fundamental advantage of a multi-species approach, with the inclusion of many taxonomic and functional groups that have a broad range of sensitivities to any given environmental regime (e.g., Frontalini et al., 2011).

The Adriatic Sea ecosystem is negatively affected by many kinds of biological and ecological threats e.g., eutrophication, pollution, fragmentation of benthic habitats, invasion of alien species (Katsanevakis et al., 2011; Pećarević et al., 2013; Corriero et al., 2016). In the regional perspective the basin is highly positioned on the list of ‘Priority issues in the Mediterranean environment’ drawn up by the European Environment Agency (EEA), with 20 (15%) out of the 131 hotspot pollution sites identified along the Mediterranean coastline in the frame of the Strategic Action Programme (SAP) of the United Nations Environment Programme (EEA & UNEP, 2006), including many harbors such as the Ancona one (Travizi et al., 2019).

Since previous works published on Ancona Harbor have always treated the different benthic components separately (Mirto & Danovaro, 2004; Spagnolo, Scarcella & Sarappa, 2011; Spagnolo et al., 2019; Baldrighi et al., 2019; Travizi et al., 2019; Franzo et al., 2022), the aim of the present work is to test whether meio- and macrofaunal assemblages could provide a comparable and/or complementary assessment of its ecological conditions.

In the present study, we characterized the meio- and macrofaunal communities (i.e., abundance, diversity, and distribution patterns) in relation to the environmental conditions they live in (i.e., concentration of selected heavy metals (HMs) and polycyclic aromatic hydrocarbons (PAHs); organic matter contents into the sediment and grain size) characterizing the Ancona Harbor. We hypothesized that the analysis of two different benthic components may give us more comprehensive information on the environmental features of the harbor, instead of a single benthic size component as is usually done (Xu, Cheung & Shin, 2014). In a future perspective, this multi-size approach should be highly recommended in coastal monitoring and management plans.

Materials and Methods

Sampling area and sampling strategy

The Ancona Harbor (water depth range, 4–15 m) is located in the western coast of central Adriatic Sea (Fig. 1), it has a water sheet of 700,000 m2 and 5,400 m of docks (Spagnolo, Scarcella & Sarappa, 2011). The harbor is one of the most important of the Adriatic Sea, with intense ferryboat and merchant ship activity. Previous investigations reported that the area is subjected to organic waste dumping derived from fishing boats and is also affected by a strong industrial pollution due to the presence of shipyards. Consequently, a huge organic matter load and high heavy metal and hydrocarbon concentrations are present inside the harbor area (Mirto & Danovaro, 2004; Bianchelli et al., 2016).

Figure 1 Map of the sampling area (Ancona Harbor) and location of the five sampling stations.

The pink rectangle indicates the geographical position of the Ancona Harbor, Italy.

In the present study, sediment samples were collected in winter 2015 from five sampling stations located from inside to outside the Ancona Harbor (Fig. 1). The five sampling stations were chosen according to the different environmental features and anthropogenic activities present in the area: the MAN station was located in the inner part of the harbor where small fishing boats dock; the PORT station was located in a transition area where work ships are berthed; the DS and LR stations were located in a more external position, nearby shipping facilities such as active berths; the API station was located outside the harbor where no activity takes place.

At each station, sediment samples for characterizing the benthic fauna (meio- and macrofauna) and environmental features were collected with a box-corer (40 cm × 30 cm wide and 50 cm high) in three independent replicates (i.e., box-corer deployments), processed and preserved differently according to the analysis to be performed (see below). At all stations, the temperature and salinity at the sea bottom were measured using CTD (Conductivity, Temperature, and Depth) probe equipped with previously calibrated sensors.

Environmental variables and microbial component

The content of each box-corer was sub-sampled with PVC corers (inner diameter, 4.5 cm) to assess the biochemical composition of the organic matter and grain size. The top 3 cm of sediment from three independent replicates for each parameter were frozen at −20 °C, except for the grain size determination, for which samples were kept at in situ temperature in single replicate until brought to the laboratory. The biochemical composition of the organic matter (total protein, carbohydrate, and lipid concentration) and chloroplastic pigments (chlorophyll-a (Chl-a) and phaeopigment (Phaeo) concentration) were determined by standard techniques (Danovaro, 2010). Concentrations were calculated using standard curves and normalized to sediment dry weight after desiccation (60 °C, 24 h). Biopolymeric organic carbon (BPC) was calculated as the sum of the carbon equivalents of carbohydrates, proteins, and lipids (Fabiano, Danovaro & Fraschetti, 1995) and was used as a proxy for the available trophic resources. The value of the protein to carbohydrate ratio (PRT/CHO) was utilized as descriptor of the nutritional quality of organic matter in the sediment, with a PRT/CHO ratio >1.0 indicating relatively high quality and high food availability (Pusceddu et al., 2010).

For grain size determination, aliquots of fresh sediment were sieved over a 63 μm mesh. The two fractions (>63 μm, sand; <63 μm, silt/mud) were dried in an oven at 60 °C and weighed. Data were expressed as a percentage of sediment total dry weight (Pusceddu et al., 2010).

Contaminants into the sediment

Concentration of selected HMs and PAHs was determined in the surface sediment (0–3 cm) collected at each sampling station and frozen at −20 °C. Analyses were conducted following previous validated methods, fully described in Benedetti et al. (2014) and Etiope et al. (2014).

In brief, HMs were determined after digestion under pressure with nitric acid and hydrogen peroxide (7:1), using a Mars 6 Microwave Digestion System (CEM Corporation, Charlotte, NC, USA). As, Cd, Cr, Cu, Fe, Mn, Ni, Pb, V, and Zn were analyzed by Atomic Absorption Spectroscopy (AAS), with flame (SpectrAA 220FS Spectrometer, Varian Inc., Palo Alto, CA, USA) and flameless atomization (240Z AA Spectrometer, Agilent Technologies Inc., Santa Clara, CA, USA), while the Hg content was quantified by Cold Vapor Atomic Absorption Spectroscopy (CVAAS; QuickTrace M-6100 Mercury Analyzer, Teledyne CETAC Technologies Ltd., Omaha, NE, USA).

PAHs were determined after KOH-methanol extraction with a Mars 6 Microwave Digestion System (CEM Corporation). Extracts were concentrated using a RC 10.09 Vacuum Concentration System (Jouan SA, Saint-Herblain, France) and purified by J.T.Baker™ BAKERBOND™ Octadecyl (C18) Solid Phase Extraction (SPE) cartridges (Avantor Inc., Radnor, PA, USA). PAHs were analyzed by High-Performance Liquid Chromatography (HPLC) using both Fluorimetric (FLD) and (UV) Diode Array Detection (DAD) (Infinity 1260 Series, Agilent Technologies, Santa Clara, CA, USA).

For both HMs and PAHs, appropriated blank solutions and the Standard Reference Material (SRM) 1944: New York/New Jersey Waterway Sediment (NIST–National Institute of Standards and Technology), digested as samples, were used to check for accuracy, precision, and recoveries of the employed analytical methodologies; concentrations obtained from SRM analyses were always within the 95% confidence intervals of the NIST certified values.

The results obtained in this study were compared with threshold values for chemicals specified in the Ministerial Decree 173/2016, the Italian normative that rules the management of dredged sediments and sets out their quality. Only those values exceeding the upper thresholds values (L2) are defined as alerting values (Table 1, Ministerial Decree 173/2016).

Table 1 Permutational multivariate analysis of variance (PERMANOVA) and pairwise test and results on total meiofaunal community composition.

In bold significant values are reported.

Permanova							
Source	d.f.	SS	MS	Pseudo-F value	P (perm)	Unique permutations	
Stations	4	7,315.80	1,829	38.0437	0.001	9,915	
Residuals	10	480.84	48.084				
Total	14	7,796.70					
Pairwise test	
Groups	t value	P (perm)	P (MC)				
MAN vs. PORT	7.39	0.103	<0.001				
MAN vs. LR	8.21	0.099	<0.001				
MAN vs. DS	4.86	0.099	0.004				
MAN vs. API	6.12	0.099	0.002				
PORT vs. LR	8.23	0.095	<0.001				
PORT vs. DS	6.44	0.101	<0.001				
PORT vs. API	5.35	0.098	0.002				
LR vs. DS	2.95	0.101	0.016				
LR vs. API	5.20	0.101	0.002				
DS vs. API	3.98	0.098	0.005				
Note:

d.f., degrees of freedom; SS, Sum of Squares; MS, Mean Square; P (perm), Permutation p-value; P (MC), Monte Carlo p-value.

Meiofauna and macrofauna

For meiofauna samples, the content of each box-corer (three independent replicates) was sub-sampled with PVC corers (inner diameter, 4.5 cm). The top 3 cm of sediment, where meiofaunal organisms are typically more abundant, were preserved in 4% buffered formaldehyde. For meiofaunal extraction, sediment samples were sieved through a 500 μm mesh; a 32 μm mesh was used to retain the smallest metazoan organisms. The latter fraction was centrifuged 3 times with LUDOX® HS-40 colloidal silica (diluted with water to a final density of 1.18 g cm−3) and stained with Rose Bengal (0.5 g L−1; Heip, Vincx & Vranken, 1985). Meiofaunal organisms were counted (no. of individuals 10 cm−2) and identified to the higher taxonomic level (i.e., Order and Class) under a Leica S8APO stereomicroscope.

For macrofauna samples, the first 20 cm of three independent box-corer deployments were sieved in situ using a 500 µm mesh and all organisms retained were preserved in 5% buffered formaldehyde. Macrofauna was sorted in laboratory using a Leica S8APO stereomicroscope and a Leica DM2500 binocular microscope, identified and classified to the lowest possible taxonomic level, and quantified (no. of individuals m−2). The data collected was subjected to a control and validation process and organized in a dedicated database. The nomenclature of the species was verified and validated using the web portal https://www.marinespecies.org/ and the ‘WoRMS Taxon Match Tool’ (WoRMS Editorial Board, 2022).

Taxon richness (i.e., Orders or Classes and species for meio- and macrofauna, respectively; S), total number of individuals per taxon/species (N), Shannon’s diversity index (H’, based on log2; Shannon & Weaver, 1949), and Pielou’s equitability index (J’; Pielou, 1969) of benthic communities were calculated.

Statistical analysis

To assess differences in benthic communities among the sampling stations a Permutational Multivariate Analysis of Variance (PERMANOVA; 9,999, number of random unrestricted permutations of raw data) was used (Anderson, 2001). The design included one factor: the sampling station (five levels, fixed). The analysis was based on Bray-Curtis’ similarity of previously fourth root transformed meio- and macrofaunal data. In case of significant differences obtained by the main test, the pairwise test was performed and, as there was a limited number of unique permutations, the p values were obtained from Monte Carlo tests (Anderson & Robinson, 2003). Permutational Multivariate Analysis of Dispersion (PERMDISP) test was applied to assess if differences among the sampling stations (between-group) were due to real differences in benthic community composition and not to differences in the multivariate dispersion of replicates (within-group) among their respective centroids. A non-metric multidimensional scaling (nMDS) ordination was carried out to visualize similarities among the sampling stations. Similarity percentages (SIMPER) analysis (cut-off, 90%) was used to identify the meio- and macrofaunal taxa that contributed to the dissimilarity among the sampling stations.

One-way analysis of variance (ANOVA) was used to explore differences among the sampling stations for organic matter content and microbial component, total abundance, diversity, and equitability indices in meio- and macrofaunal benthic communities. The ANOVA assumptions were tested graphically plotting residuals vs. fitted values, normality of residuals and residuals vs. covariate (factor station) to assess the variance homogeneity (Zuur & Ieno, 2016). When ANOVA showed significant differences, the Tukey’s honestly significant difference (HSD) test was performed to find significant effects between different levels.

Since environmental variables have been processed in single replicates, a value of 1σ (corresponding to the 70% of data around the mean in a normal standard curve; Quinn & Keough, 2002) has been subtracted and added to create the missing replicates for which the mean correspond to the original value. Two more values have been created, one lower and one higher than the actual value, and the average of the three values given exactly the observed value.

The number of environmental and pollutant-related variables were separated in two groups: pollutants (comprising HMs and PAHs) and environmental variables (comprising the organic matter compounds and silt/mud percentage). Since many compounds had linked each other, multicollinearity (cut-off at Spearman’s correlation = |0.8|) among variables was assessed to reduce the dataset and avoid problems in the analysis algorithms. In case of multicollinearity, the variables with a more stringent biological or environmental value were retained acting as proxy for the omitted collinear ones. A summary of the collinear variable and which ones have been chosen is given in Table S1. After normalization, both type of variables was used to characterize the study area by means of a principal component analysis (PCA).

Spearman’s rank correlation analysis was performed to test relationships between meio- and macrofauna total abundance, taxon richness, Shannon’s diversity (Shannon & Weaver, 1949) and Pieolu’s equitability (Pielou, 1966) indices, and the environmental features considered.

In order to verify the existence of a significant relation between the benthos data matrix and the environmental data (pollutants and environmental variables), the distance-based linear modeling (DistLM) procedure was utilized with a backward selection of the variables, and each model assessed by means of the Akaike’s Information Criterion corrected for small samples (AICc; Anderson, Gorley & Clarke, 2008). Since the environmental and pollutant variables were too numerous (even after multicollinearity assessment) respect to the meio- and macrofaunal ones, to further reduce their number, both type of variables was analyzed (separately) by means of another (PCA) to identify groups with similar variability. Only those principal components showing eigenvalues >1 (Kaiser-Guttman criterion; Zwick & Velicer, 1986) were considered. To obtain a better insight into the output loadings, the orthogonal varimax rotation of extracted PCA components was performed. After a varimax rotation, each original variable tends to be associated with one (or a small number) of PCA axis. By doing so, groups of variables were created each of which represented by a single PCA score, and all the scores obtained were used as covariates in the above DistLM analysis to assess the relations between faunal communities and the environmental features. A distance-based redundancy analysis (dbRDA) was then applied to visually investigate the relationship between the community assemblages and the environmental data (Anderson, Gorley & Clarke, 2008). All the above statistical tests were considered significant at α = 0.05. Moreover, the RELATE routine (a Mantel test like analysis) was used to search for cross-taxon correlations between meio- and macrofaunal similarity matrices.

The PERMANOVA, PERMDISP, nMDS, SIMPER, PCA, RELATE and DistLM procedures were performed with PRIMER™ and PERMANOVA+ ecological software (Clarke & Gorley, 2006; Anderson, Gorley & Clarke, 2008); the ANOVA, the Tukey’s test and the VARIMAX procedures were performed using the free software R (R Core Team, 2018; v. 4.1.3).

Results

Environmental features and contamination levels in the Ancona Harbor

Bottom water temperature reported a mean value of 10.3 °C ± 0.2 °C and salinity ranged from a minimum of 31.3 (at LR station) to a maximum of 37.6 PSU (at API station). The sandy fraction (>63 μm) characterized stations located outside the harbor (DS, LR, and API stations), while the silty muddy fraction (<63 μm) was predominant in the inner stations MAN and PORT (Table S2). Chl-a and organic matter contents into the sediments showed significantly (F4, 10 = 12.97, p = 0.001) higher values of ‘fresh’ material (i.e., Chl-a) at MAN and DS stations, while the BPC significantly (F4,10 = 22.21, p = 0.001) decreased from inside to outside the harbor (Table S2). The significant (F4,10 = 12.61, p = 0.001) highest value in the quality of organic matter (i.e. PRT/CHO) was reported at PORT station (Table S2), due to a particularly low concentration of CHO into the sediment of this station (Table S3).

Analyses aimed at determining HM concentrations in the sediment of the Ancona Harbor have highlighted a general decrease of contamination values moving from the innermost sampling station (MAN) to those one more external (Table S4). MAN station turned out to be affected by pollution levels 3–4 times higher than those detected at API station, reaching even Cu and Zn concentrations 13 and 6 times higher, respectively. Just these two elements, Cu and Zn, were the only exceeding the threshold limits imposed by the Ministerial Decree 173/2016 (46.31 µg g−1 of Cu detected at MAN station; 297.2 and 114.9 µg g−1 of Zn detected at MAN and PORT stations, respectively). In the remaining sampling stations, there were no overruns (Table S4).

The concentration levels of Σ19 PAHs ranged from 73.38 to 213.4 µg g−1 following, although not linearly, the same pattern highlighted for HMs, with higher values in the innermost sampling station (MAN), decreasing towards outer ones (Table S4). All the sediment samples showed a distinct predominance of low molecular weight PAHs, mainly driven by the Naphthalene concentration (exceeding the legislative thresholds at MAN, DS, and LR stations), which averagely accounted for 37% of the Σ19 PAHs, followed by its methylated isomers: 1- and 2-Methylnaphtalene (24% and 22%, respectively). A prevalence of volatile with 2–3 aromatic rings was reported at all stations (Fig. 2A) (for wind direction and speed on the Ancona Harbor area during February 2015, please see Fig. S1). Excluding Naphthalene and its related compounds, the PAH residual contamination was mainly ascribable to Phenanthrene and Fluoranthene (Table S4), and the principal PAH diagnostic ratios (Tobiszewski & Namieśnik, 2012), commonly used as a tool to discriminate the analyte origin and sources (Giuliani et al., 2019; Pizzini et al., 2021), indicated a petrogenic origin (Fig. 2B).

Figure 2 (A) Distribution pattern and (B) origin of polycyclic aromatic hydrocarbons (PAHs) characterizing the sediment at the investigated sampling stations.

Σ LMW = sum of low molecular weight compounds (Naphthalene, 1-Methylnaphtalene, 2-Methylnaphtalene, Acenaphthylene, Acenaphthene, Fluorene, Phenanthrene, Anthracene); Σ HMW = sum of high molecular weight compounds (Fluoranthene, Pyrene, Benz[a]anthracene, Chrysene, Benzo[b]fluoranthene, Benzo[k]fluoranthene, Benzo[a]pyrene, 7,12-Dimethylbenz[a]anthracene, Benzo[ghi]perylene, Indeno[1,2,3-cd]pyrene, Dibenz[a,h]anthracene); Σ COMB = sum of 9 non-alkylated PAHs (Fluoranthene, Pyrene, Benz[a]anthracene, Chrysene, Benzo[b]fluoranthene, Benzo[k]fluoranthene, Benzo[a]pyrene, Benzo[ghi]perylene, Indeno[1,2,3-cd]pyrene).

The results from PCA plot considering the environmental variables, summarized the differences among the sampling stations (Fig. 3). In details, MAN station clearly separated from the other stations being characterized by higher values in pollutants, food sources and mud content into the sediment. This explained the variability along the first axis (41.8% of variation). The variability along the second axis (23% of variation) was mainly explained by the contrast between the DS and LR stations and all the others, the latter characterized by the lowest values in mud content and Fluoranthene into the sediments and by intermediate values between the innermost stations and the outermost station for most of the pollutants and environmental variables considered.

Figure 3 Principal component analysis (PCA) with environmental variables.

BPC, Biopolymeric organic carbon; CPE, Chloroplastic pigment equivalent; BPER, Benzo[ghi]perylene; BAP, Benzo[a]pyrene; FLU, Fluorene; ANT/PHE, Anthracene/Phenanthrene; FLT/PYR, Fluoranthene/Pyrene.

Meiobenthic assemblages

A total of 12 taxa were identified with Nematoda representing the dominant taxon at all sampling stations, with a percentage ranging from 75% to 95% (Fig. 4A). The second most represented taxon was Copepoda with their nauplii; among less represented taxa (i.e., others) Bivalvia, Ciliata, Foraminifera, Kinorhyncha, Oligochaeta, Ostracoda, Platyhelminthes, Polychaeta, Sipuncula, and Tardigrada constituted from 3% to 15% of the meiobenthic community (Fig. 4B). Meiofauna abundance and values of its diversity indices are reported in Table S5 and represented in Fig. 5. The ANOVA tests detected significant differences among all the sampling stations for taxa richness (N; Table 2), with the highest value reported at DS station (Fig. 5A). The ANOVA tests reported significant differences among the sampling stations also for all diversity and equitability indices (Table 2 and Table S5). Regarding the number of taxa, PORT station showed a significantly lower value on average compared to all the other sampling stations (Fig. 5B). Shannon’s diversity and Pielou’s equitability indices showed both similar patterns, with MAN station presenting the highest value (Figs. 5C and 5D).

Figure 4 (A) Meiofaunal community structure and (B) contribution of taxa others at each sampling station.

Mean values of replicated samples (n = 3) are shown. Bold values reported inside the bars are total abundances (ind.10 cm−2) of meiofauna (4A) and others (4B).

Figure 5 Meiofaunal univariate measures.

(A) Meiofauna abundance (N = no. of individuals 10 cm−2) and its diversity indices: (B) meiofauna taxa richness (S), (C) Shannon’s diversity index (H’, based on log2), and (D) Pielou’s equitability index (J’). Bars represent 95% confidence intervals.

Table 2 One-way Analysis of Variance (ANOVA) results on total meiofauna and macrofauna abundance (N) and their diversity indices.

Number of taxa (S), Shannon’s diversity index (H’, based on log2), and Pielou’s equitability index (J’), characterizing the sediment at the investigated sampling stations. In bold significant p values.

Meiofauna	
Index	d.f.	F value	p value	Tukey’s HSD post hoc test	
N	Stations	4	714.30	<0.001	DS>LR>MAN>API>PORT	
Residuals	10		
S	Stations	4	11.30	<0.001	PORT<API=DS=LR=MAN	
Residuals	10		
H’	Stations	4	44.10	<0.001	MAN>PORT>API=LR(=DS)>DS	
Residuals	10				
J’	Stations	4	41.60	<0.001	MAN>PORT=API=LR(=DS)>DS	
Residuals	10				
Macrofauna	
N	Stations	4	10.60	0.002	LR>API=DS=MAN=PORT	
Residuals	10				
S	Stations	4	3.80	0.041	MAN<LR	
Residuals	10				
H’	Stations	4	8.20	0.003	LR<API=DS=MAN=PORT	
Residuals	10				
J’	Stations	4	3.90	0.036	LR<API	
Residuals	10				
Note:

d.f., degrees of freedom; HSD, Honestly Significant Difference.

PERMANOVA analysis with pairwise test reported significant differences (Table 1) in meiofaunal community composition among all the sampling stations; PERMDISP test did not show any significant dispersion around centroids, confirming that the differences among the sampling stations were due to a real difference in meiobenthic composition (Pseudo-F2,4 = 5.38, P(perm) = 0.088).

The nMDS plot (Fig. 6A) shows the separation among the sampling stations, as well as a low inter-replica variability. In particular, the innermost MAN and PORT stations were separated from to the outermost ones. SIMPER analysis detected a dissimilarity percentage from 15 (LR vs. DS stations) to 54% (PORT vs. LR stations). The highest values of dissimilarity percentage were always associated with PORT station and mainly due to very low abundances (i.e., ind. 10 cm−2 50 ± 8) in some of the most represented taxa such as: Copepoda, Foraminifera, Nematoda and Polychaeta (Table S6). For the other sampling stations, the dissimilarity was mainly due to the presence/absence or differences in the abundances of the taxa others: Bivalvia, Ciliata, Kinorhyncha, Oligochaeta, Ostracoda, Platyhelminthes and Sipuncula (Table S6). Several positive correlations emerged between meiofauna descriptors and quantity of organic matter; moreover, meiofauna abundance and its diversity were positively correlated to some HMs (i.e., Cd, Cu, Hg, Pb and Zn) and to silt/mud content into the sediment (Table S7A). The best model selected by the DistLM analysis, reported in Table 3, comprised only the pollutant compounds. The resulting dbRDA showed that the first two axes explained 83.4% of the variance of meiofaunal community composition, corresponding to the combination of the following environmental factors: V/Ni ratio (commonly used as a diagnostic marker of maritime traffic; Viana et al., 2014), Naphthalene, and Benzo[a]pyrene (ARC1) and Zn, Fluorene, and percentage of silt/mud (ARC4; Fig. S2A).

Figure 6 Non-metric multidimensional scaling (nMDS) of benthic communities.

Non-metric multidimensional scaling (nMDS) plots on (A) meiobenthic and (B) macrobenthic community structures characterizing the sediment at the investigated sampling stations. Data presented were fourth root scale transformed prior to analysis.

Table 3 Distance-based linear modeling (DistLM) analysis on meiofaunal community composition characterizing the sediment at the investigated sampling stations.

Variables are coded after Varimax Rotated PCA Axis as follow: ARCx = Abiotic variables associated to the rotated axis (x indicates the number of the axis); BRCx = Biotic variables associated to the rotated axis. For a complete list of the variables refers to the text.

Start solution	
AICc	R2	RSS	No. of variables	Selections				
80.92	0.94	447.00	6	All				
Sequential tests	
Variable	AICc	SS	Pseudo-F value	p value	Prop.	Cumul.	Res. d.f.	
BRC1	76.25	92.83	1.66	0.224	0.01	0.93	9	
BRC2	74.60	173.62	2.89	0.53	0.02	0.91	10	
Best solution	
AICc	R 2	RSS	No. of variables	Selections				
74.59	0.91	713.44	4	ARC1-ARC4				
Note:

AICc, Akaike’s Information Criterion corrected for small samples; R2 = Coefficient of determination; RSS, Residuals Sums of Squares; SS, Sums of Squares; Prop., Proportion of variation explained by the variable; Cumul., Cumulative total of Prop; Res. d.f., Residual degrees of freedom.

Macrofauna assemblages

A total of 93 taxa were identified, these included: 43 Annelida (42 Polychaeta and 1 Oligochaeta), 29 Mollusca (22 Bivalvia, 6 Gastropoda, and 1 Scaphopoda), 10 Crustacea (6 Amphipoda, 2 Cumacea, 1 Isopoda, and 1 Tanaidacea), 5 Nematoda, 2 Bryozoa, 2 Cnidaria (1 Anthozoa and 1 Hydrozoa), 1 Nemertea, and 1 Ophiuroidea (Table S8). Annelida was the most represented group (from 68% to 90% at LR and MAN stations, respectively), followed by Mollusca (from 4% to 24% at MAN and LR stations, respectively) at all the sampling stations. Other less represented groups such as Cnidaria, Isopoda, Ophiuroidea, and Tanaidacea were found only at one or two sampling stations (Fig. 7). Macrofauna abundance ranged from 610 ± 241 to 3,455 ± 425 individuals m−2 at MAN and LR stations, respectively; values of its diversity indices are reported in Table S5 are represented in Fig. 8. The ANOVA tests reported significant higher abundance value at LR station (Fig. 8A) compared to all the other stations (Table 2). LR station was also characterized by the highest number of taxa (Fig. 8B) but also by lowest values (Figs. 8C and 8D) of Shannon’s diversity and Pielou’s equitability indices (Table 2 and Table S5). PERMANOVA analysis with pairwise test reported significant differences (Table 4) in macrofaunal community composition among the majority of the sampling stations; PERMDISP test did not show any significant dispersion around centroids, confirming that the differences among the sampling stations were due to a real difference in macrobenthic composition (Pseudo-F2,4 = 0.99, P (perm) = 0.767).

Figure 7 Macrofaunal community structure at each sampling station.

Mean values of replicated samples (n = 3) are shown.

Figure 8 Macrofaunal univariate measures.

(A) Macrofauna abundance (N = no. of individuals m−2) and its diversity indices: (B) macrofauna taxa richness (S), (C) Shannon’s diversity index (H’, based on log2), and (D) Pielou’s equitability index (J’). Bars represent 95% confidence intervals.

Table 4 Permutational multivariate analysis of variance (PERMANOVA) and pairwise test and results on total macrofaunal community composition.

In bold significant values are reported.

Permanova	
Source	d.f.	SS	MS	Pseudo-F value	P (perm)	Unique permutations	
Stations	4	23,534	5,883.5	4.55	0.001	9,915	
Residuals	10	12,936	1,293.6				
Total	14	36,470					
Pairwise test	
Groups	t value	P (perm)	P (MC)				
MAN vs. PORT	2.03	0.12	0.028				
MAN vs. LR	2.67	0.10	0.011				
MAN vs. DS	2.41	0.09	0.015				
MAN vs. API	3.09	0.10	0.007				
PORT vs. LR	1.57	0.10	0.100				
PORT vs. DS	1.71	0.10	0.069				
PORT vs. API	2.03	0.10	0.031				
LR vs. DS	1.57	0.10	0.106				
LR vs. API	2.23	0.10	0.024				
DS vs. API	2.03	0.10	0.036				
Note:

d.f., degrees of freedom; SS, Sum of Squares; MS, Mean Square; P (perm), Permutation p-value; P (MC), Monte Carlo p-value.

The nMDS plot (Fig. 6B) shows the separation among the sampling stations. In detail, the innermost MAN station and for a lesser extend the outermost API station were distinguished from the others. SIMPER analysis reported high percentages of dissimilarity ranging from 63.4 (LR vs. DS stations) to 88.3% (MAN vs. API stations) between all pairs of sampling stations. The great dissimilarity is mainly due to the presence/absence of species or to a particularly abundant presence of them in one station compared to the others (Table S9). Species like Ampelisca diadema, Aponuphis bilineata, Jasmineira caudata, Spiophanes bombyx, Kurtiella bidentata, and Euclymene oerstedii characterized mainly the outermost API station, while some other species such as Tubificoides swirencoides, Streblospio sp., Heteromastus filiformis, and Chaetozone caputesocis were found inhabiting the innermost stations (Table S9). Four significant correlations (three out of four were negative) were detected between macrofauna descriptors and environmental variables; only macrofauna species richness was (negatively) correlated to three TEs and to the percentage of finest sediment fraction (Table S7B).

The best model selected by the DistLM analysis, reported in Table 5, comprised only ARC2 (Benzo[ghi]perylene, and Anthracene/(Anthracene + Phenanthrene) and Fluoranthene/(Fluoranthene + Pyrene) diagnostic ratios; Tobiszewski & Namieśnik, 2012) and ARC4 (Zn, Fluorene, and percentage of silt/mud).The resulting dbRDA showed that the first two axes explained the 42.9% of the variance in the macrofaunal community composition (Fig. S2B). This latter low percentage indicates that the residual variance associated to the community was not captured by the graph, and it is likely that an unobserved variable should have had improved the general plot. In any case, along the first axis of the dbRDA there was a clear separation between the innermost stations MAN and PORT, characterized by high values of Zn and percentage of silt/mud compared to the outermost stations LR, DS, and API; while along the second axis LR and DS stations were separated from API, MAN, and PORT stations.

Table 5 Distance-based linear modeling (DistLM) analysis on macrofaunal community composition characterizing the sediment at the investigated sampling stations.

Variables are coded after Varimax Rotated PCA Axis as follows: ARCx = Abiotic variables associated to the rotated axis (x indicates the number of the axis); BRCx = biotic variables associated to the rotated axis. For a complete list of the variables refers to the text.

Start solution	
AICc	R2	RSS	No. of variables	Selections				
28.74	0.70	11,856	6	All				
Sequential tests	
Variable	AICc	SS	Pseudo-F value	p value	Prop.	Cumul.	Res. d.f.	
BRC1	123.48	1,746.70	1.23	0.251	0.048	0.65	9	
ARC3	120.00	2,136.70	1.53	0.131	0.059	0.59	10	
BRC2	118.28	3,193.80	2.17	0.022	0.000	0.51	11	
ARC1	116.72	2,906.30	1.78	0.049	0.080	0.43	12	
Best solution	
AICc	R 2	RSS	No. of variables	Selections				
116.72	0.43	20,821	2	ARC2; ARC4				
Notes:

AICc, Akaike’s Information Criterion corrected for small samples; R2, Coefficient of determination; RSS, Residuals Sums of Squares; SS, Sums of Squares; Prop., Proportion of variation explained by the variable; Cumul., Cumulative total of Prop; Res. d.f., Residual degrees of freedom.

Discussion

Environmental features of Ancona Harbor

The presence of muddy sediments in the innermost stations and sand in the outermost ones was clearly due to a reduced exposure to hydrological factors (wind, waves, and currents) which create conditions of poor water renewal inside the Ancona Harbor, favoring the presence of fine sediments (Spagnolo, Scarcella & Sarappa, 2011). The sediment particle size can influence sediment organic matter load and pollutant content, with fine-grained components commonly showing a high content in organic matter and pollutants (Papageorgiou, Kalantzi & Karakassis, 2010). This might have facilitated an overall accumulation of HMs and PAHs inside the harbor, as reported in other harbor areas (e.g., McCready, Birch & Long, 2006; Losi et al., 2021). In particular, the high values of Cu and Zn detected at MAN and PORT stations were likely correlated to the shipyard activities present within the harbor, especially with the use of new generation antifouling paints (Costa et al., 2016; Pereira et al., 2018b). Furthermore, the pronounced prevalence of volatile, easily transportable PAHs (e.g., Naphthalene; Fig. 2A) pointed out that fuel combustion linked to maritime traffic was the major source of these organic contaminants in the harbor basin. The petrogenic origin of the PAH residual contamination (e.g., Phenanthrene and Fluoranthene; Fig. 2B) was supported by the detection in the sediment samples of V, Ni, and Pb, commonly considered as tracers of accidental oil spills and/or marine fuels (El Nemr, Khaled & El Sikaily, 2006), as well as by the values of the V/Ni ratio, marker of an intense maritime traffic (Viana et al., 2014).

Considering the threshold values for chemicals specified in the Ministerial Decree 173/2016, few values were reported exceeding the established alerting thresholds and always from the innermost stations.

Not only contaminants, but also natural and/or anthropogenic changes in the benthic trophic status (i.e., organic matter quantity and sediment biochemical composition) may affect the benthic communities (Pusceddu et al., 2011; Foti et al., 2014). The protein, carbohydrate, lipid, and BPC content in the sediments have been proposed and utilized to assess the benthic trophic status of marine coastal environments, including the Adriatic Sea (Vezzulli & Fabiano, 2006). In Dell’Anno et al. (2002) PRT and CHO sedimentary contents were suggested as proxies for organic matter quality and threshold values were established for ranking the trophic status and the environmental quality of coastal marine ecosystems. Applying those thresholds to the investigated sediments, the trophic status of Ancona Harbor could be ranked as hyper-trophic (PRT >4 mg g−1) with the exception of the API station, ranked as eutrophic (PRT ≈ 4 mg g−1). However, in terms of CHO content, Ancona Harbor should be ranked as meso-oligotrophic system (CHO <5 mg g−1). Pusceddu et al. (2009, 2011) identified as eutrophic systems those characterized by BPC concentration >3 mgC g−1, as found at MAN, PORT, LR, and DS stations, and as mesotrophic systems those characterized by BPC concentration in the range 1–3 mgC g−1, as in the case of the API station. In Ancona Harbor, the PRT/CHO ratio resulted always >1, indicating a great input of recent production’s material and highlighting the good trophic quality of the organic matter (Pusceddu et al., 2009). Increasing of organic loads in the sediment have been usually reported from other harbor areas, especially from the innermost zones (Danulat et al., 2002; Covazzi Harriague, Albertelli & Misic, 2012; Xu, Cheung & Shin, 2014; Rebai et al., 2022). Concentrations of Chl-a here reported were extremely high if compared to those reported in February in a previous study conducted along the Adriatic coasts (0.11–0.23 µg g−1; Bianchelli et al., 2016), indicating the presence of ‘fresh’ primary organic matter. However, similar results in Chl-a concentrations were reported from Tunisian harbors (Rebai et al., 2022) along with high level of organic matter. The PCA on measured environmental variables indicated the presence of a clear spatial heterogeneity among the sampling stations and a separation between innermost stations and outermost ones due to higher organic matter and contaminant loads inside the harbor basin, as previously reported in the same study area (Spagnolo, Scarcella & Sarappa, 2011; Baldrighi et al., 2019) and in other enclosed systems (Guerra-García, Corzo & García-Gómez, 2003; Vezzulli et al., 2003; Losi et al., 2013; Dauvin et al., 2017; Mehlhorn et al., 2021). This marked environmental variability in harbors is a common feature. Indeed, environmental disturbance within harbors may change rapidly over spatial scales of a few meters, depending on various factors like the localization and magnitude of pollution sources, allochthonous inputs of different nature, tidal regime, water circulation, harbor position, shape, and size (McCready, Birch & Long, 2006; Vassallo et al., 2006; Xu, Cheung & Shin, 2014).

Meiofaunal response to harbor environmental conditions

In Ancona Harbor, the meiofaunal total abundance, community structure and, for a lesser extent, univariate measures (i.e., diversity and equitability indices) reflected the marked spatial heterogeneity showed by the PCA and the clear separation both between inner and outer sampling stations and among the sampling stations themselves (MAN vs. PORT vs. DS + LR vs. API). Meiofaunal abundance was in the range of values reported by Baldrighi et al. (2019) for Ancona Harbor (i.e. from 912 ± 404 to 2,512 ± 717 ind.10 cm−2) with lower values characterizing the outermost station and for other harbors and coastal areas affected by pollution and/or high organic matter loads (Vezzulli et al., 2003; Veiga, Rubal & Besteiro, 2009; Pusceddu et al., 2011; Dal Zotto et al., 2016; Semprucci, Balsamo & Sandulli, 2016; Sedano Vera, Marquina & Espinosa Torre, 2014; Kulakova, 2022). The only exception was represented by the paucity of meiofaunal organisms found at PORT station. Considering that total meiofaunal abundance was positively linked to products derived from primary production (Chl-a, Phaeo, and chloroplast pigment equivalents−CPE), its low abundance at PORT station could be partially justified by the lowest detected value of ‘fresh’ material (Chl-a) and/or other factors known to regulate the coexistence of different organisms such as competition for resources and predation (Schratzberger et al., 2003, 2008). Moreover, given its small size, low mobility, and lack of dispersive life stages (Giere, 2009), meiofauna is more susceptible to within-habitat physical variability and environmental disturbances than larger, more mobile, and potentially more highly dispersed members of the macrofauna (Austen & Widdicombe, 2006; Schratzberger et al., 2008). This would explain the drop in meiofauna abundance, not reported for the macrofauna, at that sampling station. A total of 12 meiofauna taxa (Orders or Classes) were identified and in four sampling stations out of five the majority of taxa were represented. The measures of diversity (i.e., S, H’, and J’) were comparable to the values reported in harbor areas (e.g., Moreno et al., 2008a, 2008b; Sedano Vera, Marquina & Espinosa Torre, 2014; Kulakova, 2022) and in enclosed/transitional systems in the Adriatic Sea (e.g., Pusceddu et al., 2007, 2011; Frontalini et al., 2014; Covazzi Harriague, Albertelli & Misic, 2012). However, the strong dominance of Nematoda justified the low values reported for the Pielou’s equitability index, particularly at LR and DS stations. The dominance of the most resistant and adaptable group is a peculiarity of more stressed, less stable environments and characterized by fine sediment fraction, such as harbors (Semprucci et al., 2015). Abundance and diversity indices were correlated to different proxies of food sources (quantity and quality) into the sediment and to its grain size, confirming the effect of these environmental variables on meiofaunal populations (Balsamo et al., 2010).

The dissimilarity among the sampling stations was mainly due to some less abundant and more sensitive taxa, not present in the innermost stations. Usually, organisms that can cope with unfavorable conditions take over (e.g., Nematoda), whereas more sensitive taxa disappear or become rare (Mirto et al., 2014; Zeppilli et al., 2015). In the case of Ancona Harbor, Bivalvia, Kinorhyncha, Platyhelminthes, and Tardigrada were found only at the outermost stations (LR, DS, and API) being identified as less tolerant taxa (Baldrighi et al., 2019 and literature therein). Conversely, the more tolerant and widespread groups of Ciliata, Oligochaeta, and Polychaeta (Pusceddu et al., 2007; Moreno et al., 2008a, 2008b; Semprucci et al., 2015) characterized the innermost stations, along with a non-negligible presence (from 34% to 80% among other taxa, Fig. 4B) of soft-shelled Foraminifera inhabiting the sediment at all the investigated sampling stations. Soft-shelled monothalamous Foraminifera are an important component living in the sediment and populating the Adriatic Sea (Sabbatini et al., 2013), but most of the time this component is overlooked in meiofauna studies. The high presence of this group has been found to be associate to high values of Chl-a, eutrophic conditions, and high variability of environmental parameters (e.g., organic matter loads, salinity, temperature, oxygen content; Sabbatini et al., 2013). The strong tolerance and positive response of some monothalamous species to environmental stress (Sabbatini et al., 2010) may justify their presence in the Ancona Harbor. A further identification of the species characterizing these sediments will be necessary, in any case, to confirm our hypothesis. Changes in the community structure were supported by DistLM analysis, which revealed that pollutants and, secondly, the grain size could explain the variability in the meiofaunal composition.

Food sources did not have any effect on the meiobenthic community. According to Dell’Anno et al. (2002) and Pusceddu et al. (2009), the system of Ancona Harbor can be ranked as eutrophic (inside)—mesotrophic (outside) with high quality of organic matter. Thus, food sources did not constitute a limiting factor for the meiofaunal community (Muniz & Pires-Vanin, 2005; Covazzi Harriague, Albertelli & Misic, 2012; Dal Zotto et al., 2016), as reported instead for oligotrophic systems (e.g., Covazzi Harriague et al., 2013). Same results were reported in Franzo et al. (2022) analyzing the nematode communities inhabiting different Adriatic harbors, including that of Ancona. Authors showed that the main environmental factor that shaped the nematode assemblages in all harbors were the PAH concentration levels, while food sources and the grain size were much less relevant. Interestingly, some positive correlations between HMs and meiofaunal abundance and its related univariate measures were reported in the present study, as elsewhere (Schratzberger et al., 2003; Xu, Cheung & Shin, 2014). In the study conducted by Cibic et al. (2017), authors pinpointed as heavy metal content may influence meiofaunal abundance and its composition. The positive nature of the correlation could be the result of a meiobenthic community well adapted to permanent stress conditions (Cibic et al., 2017).

Macrofaunal response to harbor environmental conditions

In the present study, total macrofaunal abundance and its measures of diversity (i.e., S, H’, and J’), fell within the range of values reported by Spagnolo, Scarcella & Sarappa (2011) and Travizi et al. (2019) for Ancona Harbor and from other ports worldwide (e.g., Gusmao et al., 2016; Dauvin et al., 2017; Li et al., 2017; Rebai et al., 2022). Results here reported showed an overall increasing trend in macrofaunal abundance and species richness from inside to outside the study area, however, LR station significantly differed from the other sampling stations when univariate measures were considered. The low values of the Shannon’s diversity and of the Pielou’s equitability indices detected at LR station pointed to the presence of few dominant species such as Mytilus galloprovincialis and Capitella capitata. Regarding the community structure, species composition differed moving from inside to outside the harbor, overall reporting high percentages of dissimilarity among stations (SIMPER analysis, Table S9).

The macrobenthic community was mostly composed by the dominant groups of Annelida, Mollusca, and Crustacea, as usually reported from enclosed environments impacted by pollutants and characterized by high organic matter loads (Guerra-García & García-Gómez, 2004b; Spagnolo, Scarcella & Sarappa, 2011; Travizi et al., 2019). Among the group of Annelida, the Oligochaeta species T. swirencoides was identified for the first time in Ancona Harbor and it was found to be particularly abundant in all the sampling stations and even dominant at LR station. Only at API station the species was absent. Tubificid oligochaetes, also called sludge worms, are very common in high polluted areas (Brusca & Brusca, 2003) and they are recognized as a pollution-tolerant taxon (Pelletier et al., 2010). Thus, T. swirencoides was the species most responsible for the difference between the innermost sampling stations and the outermost one (SIMPER analysis, Table S9). Species composition may be affected by pollutant concentrations and high levels of organic matter, through a decrease in diversity and abundance of sensitive species (Callier et al., 2009). Most of the species found inhabiting the Ancona Harbor sediments were typically soft-bottom species and belonged to the ecological groups of disturbance-tolerant, second- and first-order opportunistic species (Borja, Franco & Perez, 2000). Polychaeta, along with the tubificid oligochaeta T. swirencoides, were the taxa most represented and diversified. Many Polychaeta species have a high level of tolerance to adverse effects such as pollution and natural perturbations (Borja, Franco & Perez, 2000), and for this reason they usually constitute the majority of benthic organisms living in harbor systems (Guerra-García & García-Gómez, 2004b). Usually, Polychaeta species richness and their diversity inside of harbor areas are low because of high pollution levels and the lack of oxygen in the water column (Estacio et al., 1997; Dhainaut-Courtois, Pruvot & Baudet, 2000). The same trend of increasing Polychaeta diversity moving outside the harbor area was also reported in the present study, with tolerant and opportunistic species such as Capitella capitata, C. caputesocis, H. filiformis, Sternaspis scutata and Streblospio sp. (Borja, Franco & Perez, 2000) particularly abundant in the innermost sampling stations. The species Prionospio cirrifera, mainly recorded outside the harbor (API station), is traditionally identified as an opportunistic spionid living in silty-clay sediments with high organic content (Borja, Franco & Perez, 2000; Simonini et al., 2004). Spagnolo, Scarcella & Sarappa (2011) reported the same finding and authors explained this as a result of a scarce tolerance of P. cirrifera to copper, detected at a concentration 9 times higher at the innermost MAN station compared to the average concentration detected at all the other sampling stations. A similar consideration could arise for the high abundance of the opportunistic species S. bombyx at API station. Polychaeta species ranked as disturbance-sensitive (Borja, Franco & Perez, 2000), such as Aricidea fragilis, Glycera capitata, Paradoneis armata, and Paraonis fulgens have also been found inhabiting the most impacted sampling stations inside the harbor but in lower abundances compared to the opportunistic species. This leads us to think about a certain kind of adaptation of these species to cope with less favorable conditions, but this aspect needs further investigation. Due to their economic and ecological importance, as well as their sedentary life, Mollusca has assumed a major role in monitoring contaminants worldwide (Pizzini et al., 2015, 2017; Grotti et al., 2016). Kurtiella bidentata, M. galloprovincialis, and Nucula nitidosa are defined as disturbance-tolerant species and they tended to dominate the innermost sampling stations in Ancona Harbor, with the only exception for K. bidentata, particularly abundant at API station. Abra alba characterized LR and DS stations, confirming its preference for sandy sediments with medium-high levels of organic matter quantity and quality (Guerra-García & García-Gómez, 2004b). This species has been reported to abound in harbors affected by heavy metal pollution (Dhainaut-Courtois, Pruvot & Baudet, 2000).

LR station was characterized by a conspicuous presence of the nonindigenous species (NIS) bivalve Anadara transversa, bivalve of Indo-Pacific origin (Streftaris & Zenetos, 2006) belonging to the family Arcidae. Members of this family have a red color blood due to a high consistent level of hemoglobin in their bodies, allowing them to colonize habitats with low oxygen concentrations (e.g., Zenetos, 1994; Morello et al., 2004).

A large number of crustaceans (Amphipoda, Isopoda, Tanaidacea) have been categorized as pollution-sensitive taxa, especially compared to Polychaeta (Pelletier et al., 2010). Crustacean communities have been considered to be among the most sensitive to changes in environmental variables (Gómez-Gesteira & Dauvin, 2000), and for this reason crustacean species richness and diversity inside harbors are generally considerably low (Estacio et al., 1997; Dhainaut-Courtois, Pruvot & Baudet, 2000). Three abundant species characterized the innermost sampling stations in Ancona Harbor: the Amphipoda Leptocheirus pilosus, the Caprellida Phtisica marina, and the Tanaidacea Apseudopsis latreillii. P. marina and A. latreillii have been reported in high number in sediments containing less sand and high concentrations of N, P, Cu, and organic matter (Guerra-García & García-Gómez, 2004a). A. latreillii belongs to the group of species that may occur under normal conditions, but whose populations are stimulated by organic enrichment, while P. marina belongs to the group of species very sensitive to organic enrichment. According to the present study and the results of other previous investigations (Conradi et al., 2000; Guerra-García & García-Gómez, 2001, 2004a) this species is able to live even in impacted habitats with moderate-high levels of heavy metals and PAHs. Conversely, the more sensitive Amphipoda Ampelisca diadema dominated the crustacean assemblages at API station. As for the meiofauna, pollutant content and the different sediment texture inside and outside the Ancona Harbor affected the macrofaunal composition, as previously reported (Guerra-García & García-Gómez, 2001, 2004a, 2004b; Spagnolo, Scarcella & Sarappa, 2011; Travizi et al., 2019). Univariate descriptors as well as the analysis of species characterizing the benthic communities indicated the presence of modified, but quite diverse and presumably well-established soft-bottom communities in all the investigated sampling stations. This might reflect the successful adaptation of many pollution-tolerant species to the long-term pollution and unstable environmental conditions of Ancona Harbor (Travizi et al., 2019).

Meiofauna and macrofauna comparison

Meio- and macrofauna can co-vary, even if the contrasting ecology of these two benthic components can give us different information on the environment where they live in (Austen, Warwick & Rosado, 1989; Schratzberger et al., 2003; Patrício et al., 2012). When compared together, both taxonomical groups inhabiting Ancona Harbor showed weak but significant relationship (Table S10), as also evident from the nMDS (Fig. 6). The weakness was likely due to the different response of meiofauna to the PORT conditions, characterized by an extreme paucity in this smaller benthic component.

However, with the only exception for PORT station, all the other stations followed the same distributions for both benthic component, notwithstanding the different intra-replica variability. This suggest that, even if the two faunal categories respond differently to the environmental and pollutant variables (or at least with different intensity), both capture and highlight the disturbances affecting the harbor area. This result, enforce the concept of cross-taxon congruence (i.e., an interoperability among groups of organisms that respond as a unique community, although both conserving their peculiarities) (Su et al., 2004). Similar congruence between meiofauna and macrofauna groups was described by, e.g., Corte et al. (2017), Cronin-O’Reilly et al. (2018). Notwithstanding, this result should be taken with caution as the relations between different groups of organisms could be weak, as it is in the present case, to provide reliable predictions of biodiversity in impacted areas (Heino, 2010), thus more data are needed to confirm or definitely discard the usability of cross-taxon congruence.

Conclusions

In considering two benthic size components at the same time (meio- and macrofauna), we were provided by a broader response to environmental conditions in the Ancona Harbor. The following considerations emerged: - The meiofauna was affected by the quality and quantity of organic matter, suggesting that meiobenthic assemblages were more receptive to within-habitat food variability than macrofauna.

- Both invertebrate groups were characterized by distinctive assemblages across the harbor, particularly evident for the meiofauna, consistent with changes detected for environmental features.

- The present investigation confirmed the fundamental advantage of a multi-benthic size approach, with the inclusion of different taxonomical groups considered to cover a broader range of functions into the ecosystem. Optimally, both groups should be used in marine pollution monitoring programs included in the EU Marine Strategy Framework Directive (MSFD, 2008; Directive 2008/56/EC) in the context of its Descriptor 1 ‘maintenance of biodiversity’ and Descriptor 6 ‘sea floor integrity’.

Supplemental Information

Supplemental Information 1 Biochemical composition of the sediment samples.

The raw data shows environmental parameters and benthic fauna measurements, as well as statistical analysis results.

Phaeo, Phaeopigment concentration; CPE, chloroplast pigment equivalents; PRT, Protein concentration; CHO, Carbohydrate concentration; LIP, Lipid concentration. Mean values ± standard deviation are reported.

Click here for additional data file.

Supplemental Information 2 Frequency percentage wind roses for (A) the sampling day and (B) the week before the sampling day.

Data retrieved from Ancona Harbor station (Latitude 43°36.579 N, Longitude 13°28.912 E) of the ISPRA (Italian Institute for Environmental Protection and Research) national tide gauge network (Rete Mareografica Nazionale).

Click here for additional data file.

Supplemental Information 3 Distance-based Redundancy Analysis (dbRDA) on benthic communities.

Distance-based Redundancy Analysis (dbRDA) graphs on (A) meiofauna and (B) macrofauna compositions, showing the sampling station distribution according to selected environmental variables. The first two axes explain (A) the 91.8% of the variability and (B) the 100% of the variability. Variables are coded after Varimax Rotated PCA Axis as follow: ARCx = Pollutant variables associated to the rotated axis (x indicates the number of the axis). For a complete list of the variables refers to the text.

Click here for additional data file.

The first author is very grateful to Dr. Jaques Grall and Vincent Le Garrec (UBO) for the long time spent on the identification of macrofaunal organisms. The authors are grateful to the crews of the boat Tecnopesca that was employed in sampling operations.

Additional Information and Declarations

Competing Interests

Author Contributions

Data Availability

Claudio Vasapollo is an Academic Editor for PeerJ.

Elisa Baldrighi conceived and designed the experiments, performed the experiments, analyzed the data, prepared figures and/or tables, authored or reviewed drafts of the article, and approved the final draft.

Sarah Pizzini analyzed the data, prepared figures and/or tables, authored or reviewed drafts of the article, and approved the final draft.

Elisa Punzo analyzed the data, authored or reviewed drafts of the article, and approved the final draft.

Angela Santelli analyzed the data, authored or reviewed drafts of the article, and approved the final draft.

Pierluigi Strafella analyzed the data, authored or reviewed drafts of the article, and approved the final draft.

Tommaso Scirocco analyzed the data, authored or reviewed drafts of the article, and approved the final draft.

Elena Manini conceived and designed the experiments, performed the experiments, analyzed the data, authored or reviewed drafts of the article, and approved the final draft.

Daniele Fattorini performed the experiments, analyzed the data, authored or reviewed drafts of the article, and approved the final draft.

Claudio Vasapollo conceived and designed the experiments, analyzed the data, prepared figures and/or tables, authored or reviewed drafts of the article, and approved the final draft.

The following information was supplied regarding data availability:

The raw measurements are available in the Supplemental Files.

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
