# Peer review of "Multi-benthic size approach to unveil different environmental conditions in a Mediterranean harbor area (Ancona, Adriatic Sea, Italy)"

_PeerJ, doi:10.7717/peerj.15541_

## Round 0.1 · original submission · Major Revisions

Dear Dr. Baldrighi,

I have received the comments of three referees that have carefully reviewed your work "Multi-benthic size approach to unveil different environmental conditions in a Mediterranean harbor area (Ancona, Adriatic Sea, Italy)." All of them believe that your work has the potential to make a relevant contribution to the field of benthic ecology; however, they also consider that major changes are needed before it could be considered for publication.

I appreciate the reviewer's effort and agree with their judgment. Therefore, although I congratulate the authors for your good work, I ask you to review the manuscript in light of the reviewers' suggestions. Please consider all reviewers' comments, including (but not restricted to) the following:

1) Analyzing and discussing meio and macrofaunal assemblages together;
2) Performing a more comprehensive review of the literature on this topic;
3) Improving the quality of some figures.

Note that all reviewers included a PDF file with annotated suggestions.

I hope you can attend to the referee's points.

Best wishes

Guilherme

·

Basic reporting

Report of Multi-benthic size approach to unveil different environmental conditions in a Mediterranean harbor area (Ancona, Adriatic Sea, Italy).
By Elisa Baldrighi et al.

The manuscript is a good contribution to the knowledge of the sedimentary environment of an important port from the commercial point of view, affected by various human activities. The medium and macro benthic communities and various environmental characteristics of the bottom sediments are studied, which are used to characterize the environmental quality of the place in a general way. What is innovative, declared by the authors, is the joint use of the two benthic communities and their potential in monitoring these environments. I understand that from the objectives declared by the authors it is intended that this "tool" is possibly of wide use (around the world) and not only for the port under study. Considering this, it is mainly that I made my contributions/suggestions and corrections.

Throughout the pdf there are multiple corrections, suggestions, comments, doubts, etc. that I understand should be addressed before the work is published.

From now on I apologize for my English, but obviously, I am not a native, I still think that the authors will have no problem understanding and if there are any doubts I am available for questions if the editor/authors think it is necessary.

In addition to everything specified in the pdf, my general commentary highlights:
The lack of a joint evaluation of the usefulness (or not) of using both communities, with their advantages and disadvantages. I think we also need to be more self-critical in relation to the fact that large groups are used for meio and the specific level for macrofauna, which will undoubtedly generate different results. This fact prevents both matrices from being treated together, and ultimately the analysis revolves around what is expected, or more expected, to occur in both communities and whether or not it is coincidental. Ultimately it would be like doing an evaluation with the meiofauan and another with the macrofauna and discussing their results in isolation.
I believe, and strongly suggest, a more joint discussion and precisely considering the strengths and weaknesses of both analyzes and of the considered strategy, being aware that it is not an analysis of both communities jointly that has been carried out.
Throughout the text I also emphasized the fact of using very regional references and not considering other geographical areas, where similar analyzes have been carried out (both communities together and separately) and perhaps making a table-level comparison with other ports around the world. , its environmental qualities, pollutants, etc., something that is more attractive thinking of a more global audience and a more global application of the results found here.

Congratulations to the authors for the work and I hope that my contributions are welcome and useful to improve its quality and that a good and attractive product for the international community can be published.
Thank you for allow me comment on the ms.

Experimental design

no more comments than those annotated in the pdf file attached

Validity of the findings

no more comments than those annotated in the pdf file attached

Additional comments

no more comments than those annotated in the pdf file attached

Reviewer 2 ·

Basic reporting

The manuscript aims to evaluate the patterns of distribution and the effect of environmental variability in the macro and meiobenthic assemblages in harbor located in the Adriatic Sea. The manuscript is generally well-written, although with some minor language revision needed. It does provide overall appropriate references, although some statements need to be reinforced with background information from previous studies. In the discussion, the references are limited in terms of scope, and very focused on comparisons with other results made in the region. The article structure is good in terms of the definition of sections; however, the way the results are organized are a little confusing, given that macro and meiofauna data are analyzed with the same methods, but results are given separately, which makes the results a little repetitive, and the manuscript with too many figures and tables. In fact, one of the main issues is that, for a manuscript that wishes to compare two assemblages, the results are very compartmentalized. This compromises the understanding of the results, as well as the discussion, as the qualitative comparisons made on the latter section end up being not very robust. In my opinion, the manuscript would benefit from a more comprehensive approach: if the objective is to compare the assemblages, then some form of analysis that encompasses both biotic components need to be considered and results have to be more integrative. Although the figures are informative, maybe some could be added as supplementary. Also, and this may be just a personal taste, some figures would benefit from an improvement in the quality and aesthetics, as there are figures that seemed to be directly taken from a standard excel spreadsheet, and aside from the low resolution, are not very pleasing in terms of design. But again, I am aware that aesthetics are a matter of taste, so the authors may take as a simple suggestion.

Experimental design

The sampling design, in technical terms, is appropriate, albeit some details need to be better explained. However, I am afraid that the number of samples is too limited to make very robust inferences. If I understood correctly, contamination and sediment parameters were grouped in a single replicate, which leave the study with 5 data points (although faunal and some variables samples seem to be replicated). However, it is not a 100% clear and some further explanation is needed. But if such is the case, it is difficult to have a robust ecological design with that few data points. For linear models, the study will end up with more variables than observations, which limit their applicability. Granted, there are methods that theoretically could address low sampling effort (see Lasso or ridge regression), but they are not addressed here and I am not sure as to whether they would be a solution in this case.
In terms of the research question, the manuscript has an hypothesis that the assemblages would have distinct responses to environmental variability, which is a valid hypothesis. However, by carrying individual analyses, with many biotic and abiotic variables and using distinct taxonomic resolutions for each assemblage, it is kinda of expected that some degree of distinctiveness would be achieved. I feel the manuscript is missing more integrative analysis, such as Procrustes or Mantel tests, that could compare assemblages. There are multiple works ( I provided some references in the attached pdf.) that aim to compare congruence among groups that could be applied here. Thus, although the statistical tools used are appropriate, some additional analysis or a distinct approach could be applied to better evaluate the assemblages concomitantly and better address the research question. However, I feel that the distinct taxonomic resolutions kinda limit the robustness of the comparisons. I understand that higher taxonomic levels may be good proxies of lower levels for some research questions, but I feel the resolution is too coarse for the meiofauna. As Nematoda is by far the most dominant taxa, its fluctuations will likely drive the changes, and the variability within this group, which could reveal distinct patterns for the meiofauna assemblage, remain hidden due to the choice of taxonomic resolution.

Validity of the findings

By the reason mentioned above, regarding the limitation of the sampling design and the compartmentalized approach to data analysis, I feel that the overall conclusions are not supported. For instance, the distinct resolution will affect the explained variance in the RDA, as you have more taxa in macrobenthic data (due to the lower resolution) and thus, more "noise" to be explained. Thus, I do not feel that the conclusion that meiofauna is more explained by a given variable is unbiased by the design choice. The abstract mentioned the complementary information the assemblages can give, but this is not, in my opinion, very strongly attached to the results. I feel that different information does not necessarily equal complementary. Also, there discussion is very centered around local studies, focusing on discussing what has been previously found and the manuscript would benefit from a more global perspective in this regard. But more importantly, the manuscript have very little in terms of discussing previous studies that directly compared meio and macrofaunal assemblages. There are some studies that have already addressed this in the marine environment. This would be important to strenghten the discussion and put the results in a wider context. For instance, the last conclusion indicates that the study points the advantages for the multi group approach, but there is very little information to support this...Is the additional effort justified for the amount of information gained? This require a much more in-depth discussion. In the current version, most of the discussion focus on harbor features and how the activities affecting the variables, comparing inner to outer areas; however, the results are not put in this context of outer and inner areas and the number of samples again seem to limit inferences regarding the spatial patterns.

Additional comments

Most of my concerns were based on some questions regarding the methods, such as the exact number of samples and the taxonomic resolution used. I feel that those are issues that can be better clarified or further detailed. Granted, major changes would be required to address some of the perceived issues. But if the authors are willing to carry out this effort and feel that the points raised can be appropriately addressed, I believe the manuscript could be a proper addition to PeerJ, given that the questions are important and this kind of comparative study, although not a novelty, are certainly welcome. I am aware of the hardships to work with one of the assemblages, so it is commendable that the authors aimed to tackle both at once.

I have made additional and specific comments in the attached pdf. I hope the authors find the suggestions useful. Please, be aware that I write too many comments, but much of the content are small changes, better detailing or just general observations, and the goal was to help to the extent of my capability. Of course, I do not expect authors to agree with all suggestions; I may be wrong in some of my assessments, but I ask to please provide proper justifications in these instances.

Annotated reviews are not available for download in order to protect the identity of reviewers who chose to remain anonymous.

Reviewer 3 ·

Basic reporting

This study represents an important and interesting attempt to document the differential response of meiofauna and macrofauna in port systems. Overall the ms is original, well-written and based on a solid data set. The methods and statistical analyses are appropriate, but I have some doubts on the interpretation of some results and I suggest to the authors a futher check of some issues that I reported in the pdf file. As such I can recommend the publication of this Ms with some major revisions that I am sure that the authors can easily address.

Experimental design

The experimental design seems rigorous and all the information are available in the methods sections

Validity of the findings

As I reported above, I think that statistical approach is fine, but there are some interpretations that should be carefully checked

Annotated reviews are not available for download in order to protect the identity of reviewers who chose to remain anonymous.

---

## Round 0.2 · accepted · Accept

Dear Dr. Baldrighi,

The reviewers are satisfied with your revision, and I thank you for addressing all their comments thoroughly.

Congratulations on your great work!

Best regards

Guilherme

·

Basic reporting

Dear Editor
The authors have done an excellent reviewing and improving of the MS in question. I see in this new version that most of the corrections and suggestions made by the three reviewers were addressed and in my opinion the ms would now be able to be published, I think it will be a valuable contribution to the area of marine ecology and fit very well with the scope of the journal.
Thanks for your invitation,
Pablo Muniz

Experimental design

ok

Validity of the findings

ok

Additional comments

ok

Reviewer 3 ·

Basic reporting

The manuscript is now well written, and much improved in comparison with the previous version

Experimental design

Methods and approches were well detailed

Validity of the findings

The findings of this paper add novelty in the field and could allow a good comprension of the effects of human activities in the marine systems